# Influence-Based Mini-Batching for Graph Neural Networks

**Johannes Gasteiger**[*]
Technical University of Munich[†]
johannes.gasteiger@tum.de

**Chendi Qian**[*]
Technical University of Munich
chendi.qian@tum.de

**Stephan Günnemann**
Technical University of Munich
stephan.guennemann@tum.de

## Abstract

Using graph neural networks for large graphs is challenging since there is no clear way of constructing mini-batches. To solve this, previous methods have relied on sampling or graph clustering. While these approaches often lead to good training convergence, they introduce significant overhead due to expensive random data accesses and perform poorly during inference. In this work we instead focus on model behavior during inference. We theoretically model batch construction via maximizing the influence score of nodes on the outputs. This formulation leads to optimal approximation of the output when we do not have knowledge of the trained model. We call the resulting method influence-based mini-batching (IBMB). IBMB accelerates inference by up to 130x compared to previous methods that reach similar accuracy. Remarkably, with adaptive optimization and the right training schedule IBMB can also substantially accelerate training, thanks to precomputed batches and consecutive memory accesses. This results in up to 18x faster training per epoch and up to 17x faster convergence per runtime compared to previous methods.

## 1 Introduction

Creating mini-batches is highly non-trivial for connected data, since it requires selecting a meaningful subset despite the data's connectedness. When the graph does not fit into memory, the mini-batching problem is equally relevant for both inference and training. However, mini-batching methods have so far mostly been focused on training, despite the major practical importance of inference. Once a model is put into production, it continuously runs inference to serve user queries. On AWS, more than 90 % of infrastructure cost is due to inference, and less than 10 % is due to training [24]. Even during training, inference is necessary for early stopping and performance monitoring. A training method thus has rather limited utility by itself.

Selecting mini-batches for inference is distinctly different from training. Instead of averaging out stochastic sampling effects over many training steps, we need to ensure that every prediction is as accurate as possible. To achieve this, we propose a theoretical framework for creating mini-batches based on the expected influence of nodes on the outputs. Selecting nodes according to this formulation provably leads to an optimal approximation of the output. The resulting optimization problem shows that we need to distinguish between two classes of nodes: Output nodes and auxiliary nodes. Output nodes are those for which we compute a prediction *in this batch*, for example a set of validation nodes. Auxiliary nodes provide inputs and define the batch's subgraph. This distinction allows us to choose a meaningful neighborhood for every prediction, while ignoring irrelevant parts of the graph. Note that output nodes in one batch can be auxiliary nodes in another batch.

This distinction furthermore splits mini-batching into two problems: 1. How do we partition output nodes into efficient mini-batches? 2. How do we choose the auxiliary nodes for a given set of output nodes? Having split the problem like this, we see that most previous works either focus exclusively on the first question by only using graph partitions [7] or on the second question and choose a uniformly random subset of nodes as output nodes [21, 42]. Jointly considering both aspects with an overarching

---

[*]Equal contribution.
[†]Now at Google Research.

Gasteiger, Qian, Günnemann, Influence-Based Mini-Batching for Graph Neural Networks. *Proceedings of the First Learning on Graphs Conference (LoG 2022)*, PMLR 198, Virtual Event, December 9–12, 2022.

theoretical framework allows for substantial synergy effects. For example, batching nearby output nodes together allows one output node to leverage another one's auxiliary nodes.

We call this overall framework influence-based mini-batching (IBMB). On the practical side, we propose two instantiations of IBMB by approximating the influence between nodes via personalized PageRank (PPR). We use fast approximations of PPR to select auxiliary nodes by their highest PPR scores. Accordingly, we partition output nodes using PPR-based node distances or via graph partitioning. We then use the subgraph induced by these nodes as a mini-batch. IBMB accelerates inference by up to 130x compared to previous methods that achieve similar accuracy.

Remarkably, we found that IBMB also works well for training, despite being derived from inference. This is due to the computational advantage of precomputed mini-batches, which can be loaded from a cache to ensure efficient memory accesses. We counteract the negative effect of the resulting sparse mini-batch gradients via adaptive optimization and batch scheduling. Overall, IBMB achieves an up to 18x improvement in time per training epoch, with similar final accuracy. This fast runtime more than makes up for any slow-down in convergence per step. Its speed advantage grows even further for the common setting of low label ratios, since our method avoids computation on irrelevant parts of the graph. Our implementation is available online[3]. In summary, our core contributions are:

- Influence-based mini-batching (IBMB): A theoretical framework for selecting mini-batches for GNN inference based on influence scores.
- Practical instantiations of IBMB that work for a variety of GNNs and datasets. They substantially accelerate inference and training without sacrificing accuracy, especially for small label ratios.
- Methods for mitigating the impact of fixed, local mini-batches and sparse gradients on training.

## 2 Background and related work

**Graph neural networks.** We consider a graph $\mathcal{G} = (\mathcal{V}, \mathcal{E})$ with node set $\mathcal{V}$ and (possibly directed) edge set $\mathcal{E}$. $N = |\mathcal{V}|$ denotes the number of nodes, $E = |\mathcal{E}|$ the number of edges, and $\boldsymbol{A} \in \mathbb{R}^{N \times N}$ the adjacency matrix. GNNs use one embedding per node $\boldsymbol{h}_u \in \mathbb{R}^H$ and edge $\boldsymbol{e}_{(uv)} \in \mathbb{R}^{H_e}$ of size $H$ and $H_e$, and update them in each layer via message passing between neighboring nodes. We denote the embedding in layer $l$ as $\boldsymbol{h}_u^{(l)}$ and its $i$'th entry as $h_{ui}^{(l)}$. Most GNNs can be expressed via the following equations:

$$\boldsymbol{h}_u^{(l+1)} = f_{\text{node}}(\boldsymbol{h}_u^{(l)}, \underset{v \in \mathcal{N}_u}{\text{Agg}} [f_{\text{msg}}(\boldsymbol{h}_u^{(l)}, \boldsymbol{h}_v^{(l)}, \boldsymbol{e}_{(uv)}^{(l)})]), \tag{1}$$

$$\boldsymbol{e}_{(uv)}^{(l+1)} = f_{\text{edge}}(\boldsymbol{h}_u^{(l+1)}, \boldsymbol{h}_v^{(l+1)}, \boldsymbol{e}_{(uv)}^{(l)}). \tag{2}$$

The node and edge update functions $f_{\text{node}}$ and $f_{\text{edge}}$, and the message function $f_{\text{msg}}$ can be implemented using e.g. linear layers, multi-layer perceptrons (MLPs), and skip connections. The node's neighborhood $\mathcal{N}_u$ is usually defined directly by the graph $\mathcal{G}$ [27], but can be generalized to consider larger or even global neighborhoods [1, 16], or feature similarity [10]. The most common aggregation function Agg is summation, but multiple other alternatives have also been explored [9, 17]. Edge embeddings $\boldsymbol{e}_{(uv)}$ are often not used in GNNs, but some variants rely on them exclusively [6].

**Scalable GNNs.** Multiple works have proposed massively scalable GNNs that leverage the peculiarities of message passing to condense it into a single step, akin to label or feature propagation [4, 14]. Our work focuses on general, model-agnostic scalability methods.

**Scalable graph learning.** Classical graph learning faced issues similar to GNNs when scaling to large graphs. Multiple frameworks for distributed graph computations were proposed to solve this without approximations or sampling [19, 28, 31, 32]. Other works scaled to large graphs via stochastic variational inference, e.g. by sampling nodes and node pairs [20]. Interestingly, this approach is quite similar to sampling-based mini-batching for GNNs.

**Mini batching for GNNs.** Previous mini-batching methods can largely be divided into three categories: Node-wise sampling, layer-wise sampling, and subgraph-based sampling [29]. In node-wise sampling, we obtain a separate set of auxiliary nodes for every output node, which are sampled independently for each message passing step. Each output node is treated independently; if two output nodes sample the same auxiliary node, we compute its embedding twice [21, 30, 39]. Layer-wise

---
[3]https://www.cs.cit.tum.de/daml/ibmb

sampling jointly considers all output nodes of a batch to compute a stochastic set of activations in each layer. Computations on auxiliary nodes are thus shared [5, 23, 42]. Subgraph-based sampling selects a meaningful subgraph and then runs the GNN on this subgraph as if it were the full graph. This method thus computes the outputs and intermediate embeddings of all nodes in that subgraph [7, 40]. Our method most closely resembles the subgraph-based sampling approach. However, IBMB considers both output and auxiliary nodes, resulting in better batches, and only computes the output of predetermined output nodes, similar to node-wise sampling. Two particularly related methods are Cluster-GCN [7] and shaDow [41]. However, both of these methods cover only part of the IBMB framework. Cluster-GCN resembles the graph partitioning variant of output node partitioning (see Sec. 3.2). It does not select the most relevant auxiliary nodes and cannot ignore irrelevant parts of the graph. IBMB thus exhibits better accuracy for output nodes close to the partition boundary and is significantly faster when training on small training sets. ShaDow is related to IBMB's node-wise auxiliary node selection variant (see Sec. 3.1). ShaDow does not address output node partitioning, resulting in worse runtimes and inconsistent accuracy in both inference and training.

Note that mini-batch generation is an orthogonal problem to training frameworks such as GNNAutoScale [13]. We can use IBMB to provide mini-batches as part of GNNAutoScale.

## 3 Influence-based mini-batching

**Influence scores.** To effectively create graph-based mini-batches we must first quantify how important one node is for another node's prediction. As proposed by Xu et al. [38], we can do this via the influence score, which determines the local sensitivity of the output at node $u$ on the input at node $v$ as:

$$I(v, u) = \sum_i \sum_j \left| \frac{\partial \boldsymbol{h}_{ui}^{(L)}}{\partial \boldsymbol{X}_{vj}} \right|, \tag{3}$$

where $\boldsymbol{h}_{ui}^{(L)}$ is the $i$'th entry in the embedding of node $u$ in the last layer $L$ and $\boldsymbol{X}_{vj}$ is feature $j$ of node $v$. Analyzing the expected influence score can provide a crisp understanding of how to select nodes for inference when we only have knowledge of the graph, not the model or the node features. To formally prove this connection, we consider a slightly limited class of GNNs and model our lack of knowledge via a randomization assumption of ReLU activations, similar to Choromanska et al. [8], and by assuming that all nodes have the same expected features, yielding (proof in App. A):

**Theorem 1.** *Given a GNN with linear, graph-dependent aggregation and ReLU activations. Assume that all paths in the model's computation graph are activated with the same probability $\rho$ and nodes have features with expected value $\mathbb{E}[X_{v,i}] = \chi_i$. If we restrict the model input features to a set of auxiliary nodes $\mathcal{S}_{\mathrm{aux}} \subseteq \mathcal{V}$, then the error*

$$\|\tilde{\boldsymbol{h}}_u^{(L)} - \boldsymbol{h}_u^{(L)}\|_1 \tag{4}$$

*between the approximate logits $\tilde{\boldsymbol{h}}_u^{(L)}$ and the true logits $\boldsymbol{h}_u^{(L)}$ is minimized, in expectation, by selecting the nodes $v \in \mathcal{S}_{\mathrm{aux}}$ with maximum influence score $I(v, u)$.*

**Formalizing mini-batching.** We can leverage this insight by formalizing mini-batching as the optimization problem

$$\max_{\substack{P_{\mathrm{out}} \in \mathbb{P}(\mathcal{V}_{\mathrm{out}}) \\ |P_{\mathrm{out}}| = b}} \sum_{\mathcal{S}_{\mathrm{out}} \in P_{\mathrm{out}}} \underbrace{\max_{\substack{\mathcal{S}_{\mathrm{aux}} \subseteq \mathcal{V} \\ |\mathcal{S}_{\mathrm{aux}}| \leq B}} \sum_{u \in \mathcal{S}_{\mathrm{out}}} \sum_{v \in \mathcal{S}_{\mathrm{aux}}}}_{} \underbrace{I(v, u)}_{}, \tag{5}$$

$$\underbrace{\phantom{\max_{\substack{P_{\mathrm{out}} \in \mathbb{P}(\mathcal{V}_{\mathrm{out}}) \\ |P_{\mathrm{out}}| = b}}}}_{\text{Output node partitioning}} \underbrace{\phantom{\max_{\substack{\mathcal{S}_{\mathrm{aux}} \subseteq \mathcal{V} \\ |\mathcal{S}_{\mathrm{aux}}| \leq B}}}}_{\text{Auxiliary node selection}} \underbrace{\phantom{I(v,u)}}_{\text{Influence score}}$$

where $\mathbb{P}(\mathcal{V}_{\mathrm{out}})$ denotes the set of partitions of the output nodes $\mathcal{V}_{\mathrm{out}}$, $b$ the number of batches, and $B$ the maximum batch size. This optimization yields two results: The output node partition $P_{\mathrm{out}}$ and the auxiliary node set for each batch of output nodes, $\mathcal{S}_{\mathrm{aux}}$. The hyperparameter $B$ is determined by the available (GPU) memory, while $b$ trades off runtime and approximation quality. This formulation optimizes the average approximation across all outputs. This might not be ideal since some nodes might already be approximated well with a lower number of auxiliary nodes. We can instead focus on the worst-case approximation by optimizing the minimum aggregate influence score as

$$\max_{\substack{P_{\mathrm{out}} \in \mathbb{P}(\mathcal{V}_{\mathrm{out}}) \\ |P_{\mathrm{out}}| = b}} \min_{\mathcal{S}_{\mathrm{out}} \in P_{\mathrm{out}}} \underbrace{\max_{\substack{\mathcal{S}_{\mathrm{aux}} \subseteq \mathcal{V} \\ |\mathcal{S}_{\mathrm{aux}}| \leq B}} \min_{u \in \mathcal{S}_{\mathrm{out}}} \sum_{v \in \mathcal{S}_{\mathrm{aux}}}}_{} \underbrace{I(v, u)}_{}. \tag{6}$$

$$\underbrace{\phantom{\max \min}}_{\text{Output node partitioning}} \underbrace{\phantom{\max \min}}_{\text{Auxiliary node selection}} \underbrace{\phantom{I(v,u)}}_{\text{Influence score}}$$

Both Eqs. (5) and (6) split the mini-batching problem into three parts: Output node partitioning, auxiliary node selection, and influence score computation. We call this approach influence-based mini-batching (IBMB).

**Computing influence scores.** The model's influence score depends on various model details, especially when considering exact, trained models. In many cases we can calculate the expected influence score by making simplifying assumptions, similar to Theorem 1. This allows tailoring the mini-batching method to the exact model of interest. For the remainder of this work we will focus our analysis on the broad class of models that use the average as an aggregation function, such as graph convolutional networks (GCN) [27]. In this case, we can make similar assumptions on the GNN as in Theorem 1 to prove that the influence score is proportional to a slightly modified random walk with $L$ steps [38]. To remove the influence score's dependence on the number of layers $L$, we can furthermore take the limit $L \to \infty$. Unfortunately, this would result in a limit distribution that is independent of node $v$. To avoid this we add restarts to the random walk, as proposed by Gasteiger et al. [15]. The limit $L \to \infty$ then becomes equivalent to personalized PageRank (PPR), which we can thus use an approximation of the influence score. Notably, PPR even works well for models with more complex, data-dependent influence scores, such as GAT (see Sec. 5). The PPR matrix is given by

$$\mathbf{\Pi}^{\mathrm{ppr}} = \alpha(\boldsymbol{I}_N - (1-\alpha)\boldsymbol{D}^{-1}\boldsymbol{A})^{-1}, \tag{7}$$

with the teleport probability $\alpha \in (0, 1]$ and the diagonal degree matrix $\boldsymbol{D}_{ii} = \sum_k \boldsymbol{A}_{ik}$. The entry $\mathbf{\Pi}^{\mathrm{ppr}}_{uv}$ then provides a measure for the influence of node $v$ on $u$. Calculating the above inverse is obviously infeasible for large graphs. However, we can approximate $\mathbf{\Pi}^{\mathrm{ppr}}$ with a sparse matrix $\tilde{\mathbf{\Pi}}^{\mathrm{ppr}}$ in time $\mathcal{O}(\frac{1}{\varepsilon\alpha})$ per row, with error $\varepsilon \deg(v)$ [2]. Importantly, this approximation uses only the node's local neighborhood, making its runtime independent of the overall graph size and thus massively scalable. Furthermore, the calculation is deterministic and model-independent, so we only need to perform this computation once during preprocessing.

## 3.1 Auxiliary node selection

**Node-wise selection.** Selecting auxiliary nodes on large graphs requires a method that efficiently yields nodes with highest expected influence. Fortunately, there is a well-developed literature of methods for finding the top-k PPR nodes. The classic approximate PPR method [2] is guaranteed to provide all nodes with a PPR value $\mathbf{\Pi}^{\mathrm{ppr}}_{uv} > \varepsilon \deg(v)$ w.r.t. the root (output) node $u$. Optimizing auxiliary nodes by the worst-case influence score (Eq. (6)) thus equates to separately running approximate PPR for each output node in a batch $\mathcal{S}_{\mathrm{out}}$, and then merging them.

**Batch-wise selection.** Considering each output node separately does not take into account how one auxiliary node jointly affects multiple output nodes, as required for the average-case formulation in Eq. (5). Fortunately, PPR calculation can be adapted to use a set of root nodes. To do so, we use a set of nodes in the teleport vector $\boldsymbol{t}$ instead of a single node, e.g. by leveraging the underlying recursive equation for a PPR vector $\pi_{\mathrm{ppr}}(\boldsymbol{t}) = (1-\alpha)\boldsymbol{D}^{-1}\boldsymbol{A}\pi_{\mathrm{ppr}}(\boldsymbol{t}) + \alpha\boldsymbol{t}$. $\boldsymbol{t}$ is a one-hot vector in the node-wise setting, while for batch-wise PPR it is $1/|\mathcal{S}_{\mathrm{out}}|$ for all nodes in $\mathcal{S}_{\mathrm{out}}$. This variant is also known as topic-sensitive PageRank. We found that batch-wise PPR is significantly faster than node-wise PPR. However, it can lead to cases where one outlier node receives almost no neighbors, while others have excessively many. Whether node-wise or batch-wise selection performs better thus often depends on the dataset and model.

**Subgraph generation.** Creating mini-batches also requires selecting a subgraph of relevant edges. We do so by using the subgraph induced by the selected output and auxiliary nodes in a batch. Note that the above node selection methods ignore how these changes to the graph affect the influence scores. This is a limitation of these methods. However, PPR is a *local clustering* method and we can thus expect auxiliary nodes to be well-connected.

## 3.2 Output node partitioning

**Optimal partitioning.** Finding the optimal node partition in Eqs. (5) and (6) would require trying out every possible partition since a change in $\mathcal{S}_{\mathrm{out}}$ can unpredictably affect the optimal choice of auxiliary nodes. Doing so is clearly intractable since the number of partitions grows exponentially with $N$ for a fixed batch size. We thus need to approximate the optimal partition via a scalable heuristic. The implicit goal of this step is finding output nodes that share a large number of auxiliary nodes. One good proxy for these overlaps is the proximity of nodes in the graph.

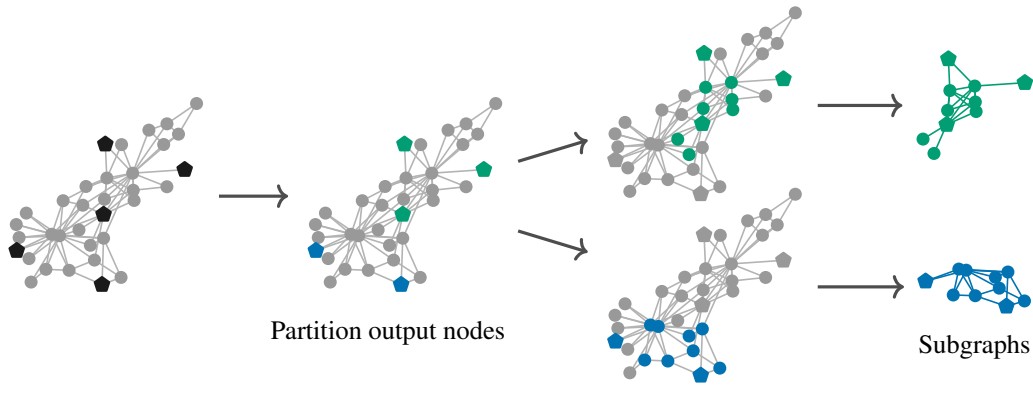

Partition output nodes

Select auxiliary nodes

Subgraphs

**Figure 1:** Practical example of influence-based mini-batching (IBMB). The output nodes are indicated by pentagons. These nodes are first partitioned into batches, e.g. by grouping nearby nodes together. We then use influence scores to select the auxiliary nodes of each batch, e.g. neighbors with top-$k$ personalized PageRank (PPR) scores. Finally, we generate a batch using the induced subgraph of all selected nodes, but only calculate the outputs of the output nodes we chose when partitioning. Batches can overlap and do not need to cover the whole graph.

**Distance-based partitioning.** We propose two methods that leverage graph locality as a heuristic. The first is based on node distances. In this approach we first compute the pairwise node distances between nodes that are close in the graph. We can use PPR for this as well, since it is also commonly used as a node distance. If we select auxiliary nodes with node-wise PPR, we thus only need to calculate PPR scores once for both steps.

Next, we greedily construct the partition $P_{\text{out}}$ from $\tilde{\mathbf{\Pi}}^{\text{ppr}}$. To do so, we start by putting every node $u$ into a separate batch $\{u\}$. We then sort all elements in $\tilde{\mathbf{\Pi}}^{\text{ppr}}$ by magnitude, independent of their row or column. We scan over these values in descending order, considering the value's indices $(u, v)$ and merging the batches containing the two nodes. Afterwards we randomly merge any small leftover batches. We stay within memory constraints by only merging batches that stay below the maximum batch size $B$. This method achieves well-overlapping batches and can efficiently add incrementally incoming out nodes, e.g. in a streaming setting. Our experiments show that this method achieves a good compromise between well-overlapping batches and good gradients for training (see Sec. 5). Note that the resulting partition is unbalanced, i.e. some sets will be larger than others.

**Graph partitioning.** For our second method, we note that partitioning output nodes into overlapping mini-batches is closely related to partitioning graphs. We can thus leverage the extensive research on this topic by using the METIS graph partitioning algorithm [25] to find a partition of output nodes $P_{\text{out}}$. We found that graph partitioning yields roughly a two times higher overlap of auxiliary nodes than distance-based partitioning, thus leading to significantly more efficient batches. However, it also results in worse gradient samples, which we found to be detrimental for training (see Sec. 5).

**Computational complexity.** Since IBMB ignores irrelevant parts of the graph, inference and training scale linearly in the number of output nodes $\mathcal{O}(N_{\text{out}})$. Preprocessing runs in $\mathcal{O}(\frac{N_{\text{out}}}{\epsilon\alpha})$ for node-wise PPR-based steps, $\mathcal{O}(\frac{b}{\epsilon\alpha})$ for batch-wise PPR, and in $\mathcal{O}(E)$ for graph partitioning. The runtime of IBMB is thus *independent* of the graph size if we use distance-based partitioning. Fig. 1 gives an overview of the full practical IBMB process.

## 4 Training with IBMB

**Computational advantages.** The above analysis focused on node outputs, not gradient estimation and training. However, IBMB also has inherent advantages for training, since we need to perform mini-batch generation only once during preprocessing. We can then cache each mini-batch in consecutive blocks of memory, thereby allowing the data to be stored where it is needed and circumventing

expensive random data accesses. This significantly accelerates training, allows efficient distributed training, and enables more expensive node selection procedures. In contrast, most previous methods select both output and auxiliary nodes randomly in each epoch, which incurs significant overhead. Our experiments show that IBMB's more efficient memory accesses clearly outweigh the slightly worse gradient estimates (see Sec. 5). This seems counter-intuitive since the deterministic, fixed mini-batches in IBMB only provide sparse, fixed gradient samples. In this section we discuss these aspects and how adaptive optimization and batch scheduling counteract their effects.

**Sparse gradients.** Partitioning output nodes based on proximity effectively correlates the gradients sampled in a batch. The model thus sees a sparse gradient sample, which does not cover all aspects of the dataset. Fortunately, adaptive optimization methods such as Adagrad and Adam were developed exactly for such sparse gradients [12, 26]. We furthermore ensure an unbiased training process by using every output (training) node exactly once per epoch.

**Fixed batches.** Using a *fixed* set of batches can lead to problems with basic stochastic gradient descend (SGD) as well. Imagine training with two fixed batches whose loss functions have different minima. If training has "converged" to one of these minima, SGD would start to oscillate: It would take one step towards the other minimum, and then back, and so forth. To counteract this oscillation, we could add a "consensus constraint" to enforce a consensus between the weights after different batches, akin to distributed optimization [33]. We can solve this constraint using a primal-dual saddle-point algorithm with directed communication [18]. The resulting dynamics are $\dot{x}^{(t)} = -\nabla \tilde{f}^{(t)}(x^{(t)}) - \alpha\lambda\dot{x}^{(t-1)} - \lambda^2\dot{x}^{(t-2)}$, with the weights $x^{(t)}$ at time step $t$, the learning rate $\lambda$ and the dual variable $\alpha$. These dynamics resemble SGD with momentum, and fit perfectly into the framework of adaptive optimization methods [34]. Indeed, momentum and adaptive methods suppress the oscillations in the above example with two minima. Accordingly, prior works have also found benefits in deterministically selecting fixed mini-batches [3, 37]. We further improve convergence by adaptively reducing the learning rate when the validation loss plateaus, which ensures that the step size decreases consistently.

**Batch scheduling.** While Adam with learning rate scheduling consistently ensures convergence, we still observe downward spikes in accuracy during training. To illustrate this issue, consider a sequence of mini-batches. In regular training every mini-batch is similar and the order of these batches is irrelevant. In our case, however, some of the mini-batches might be very similar. If the optimizer sees a series of similar batches, it will take increasingly large steps in a suboptimal direction, which leads to the observed downward spikes in accuracy. We propose to prevent these suboptimal batch sequences by optimizing the order of batches. To quantify batch similarity we measure the symmetrized KL-divergence of the label distribution between batches. In particular, we use the normalized training label distribution $p_i = c_i / \sum_j c_j$, where $c_i$ is the number of training nodes of class $i$. This results in the pairwise batch distance $d_{ab}$ between batches $a$ and $b$. We propose two ways to use this for improving the batch schedule: (i) Find the fixed batch cycle that *maximizes* the batch distances between consecutive batches. This is a traveling salesman problem for finding the maximum distance loop that visits all batches. It is therefore only feasible for a small number of batches. (ii) Sample the next batch weighted by the distance to the current batch. Both scheduling methods improve convergence and increase final accuracy, at almost no cost during training. Overall, our training scheme leads to consistent convergence. Even accumulating gradients over the whole epoch does not significantly change convergence or final accuracy (see Fig. 8).

## 5 Experiments

**Experimental setup.** We show results for two variants of our method: IBMB with PPR distance-based batches and node-wise PPR clustering (node-wise IBMB), and IBMB with graph partition-based batches and batch-wise PPR clustering (batch-wise IBMB). We also experimented with the two other combinations of the output node partitioning and auxiliary node selection variants, but found these two to work best. We compare them to four state-of-the-art mini-batching methods: Neighbor sampling [21], Layer-Dependent Importance Sampling (LADIES) [42], GraphSAINT-RW [40], shaDow [41], and Cluster-GCN [7]. We use four large node classification datasets for evaluation: ogbn-arxiv [22, 36, ODC-BY], ogbn-products [36, Amazon license], Reddit [21], and ogbn-papers100M [22, 36, ODC-BY]. While these datasets use the transductive setting, IBMB makes no assumptions about this and can equally be applied to the inductive setting. We skip the common, small datasets (Cora, Citeseer, PubMed) since they are ill-suited for evaluating scalability methods. We do not strive to set a new accu-

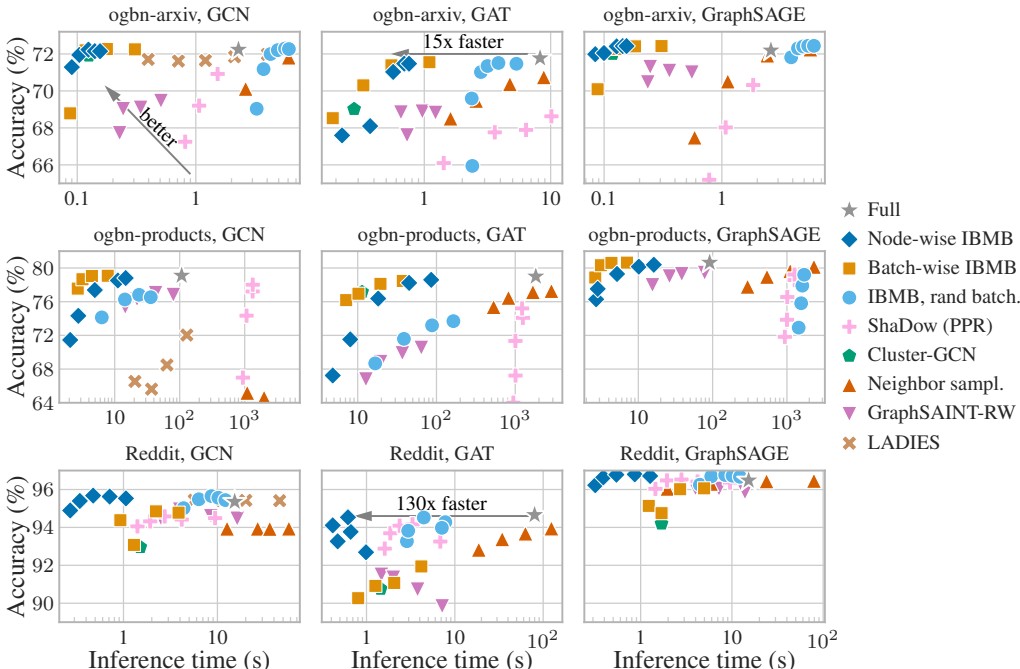

**Figure 2:** Test accuracy and log. inference time averaged over a fixed set of 10 pretrained GNNs. IBMB provides the best accuracy versus time trade-off (top-left corner) in all settings.

racy record but instead aim for a consistent, fair comparison based on three standard GNNs: GCN [27], graph attention networks (GAT) [35], and GraphSAGE [21]. We use the same training pipeline for all methods, giving them access to the same optimizations. Since full inference is too slow to execute every epoch we use the mini-batching method used for training to also approximate inference during training. We run each experiment 10 times and report the mean and standard deviation in all tables and the bootstrapped mean and 95 % confidence intervals in all figures. We fully pipeline data loading and batch creation by prefetching batches in parallel. We found that using more than one worker for this does not improve runtime, most likely because data loading is limited by memory bandwidth, which is shared between workers. We keep GPU memory usage constant between methods, and tune all remaining hyperparameters for both IBMB and the baselines. See App. B for full experimental details.

**Inference.** Fig. 2 compares inference accuracy and time of different batching methods, using the same pretrained model and varying computational budgets (number of auxiliary nodes/sampled nodes) at a fixed GPU memory budget. IBMB provides the best trade-off between accuracy and time in all settings. Node-wise IBMB performs better than graph partitioning, except on ogbn-products. IBMB provides a significant speedup over chunking-based full-batch inference on GPU, being 10 to 900 times faster at comparable accuracy. All previous methods are either significantly slower or less accurate.

**Training.** IBMB performs significantly better in training than previous methods, converging up to 17x faster than all baselines (see Fig. 3). This is *despite* the fact that we always prefetch the next batch in parallel. Note that GAT is slower to compute than GCN and GraphSAGE, limiting the positive impact of a fast batching method. Compute-constrained models like GAT are less relevant in practice since data access is typically the bottleneck for GNNs on large, often even disk-based datasets [4]. Table 7 in the appendix furthermore shows that IBMB's time per epoch is significantly faster than all sampling-based methods. Cluster-GCN has a comparable runtime, which is expected due to its similarity with IBMB. However, it converges more slowly than IBMB and reaches substantially lower final accuracy. Neighbor sampling achieves good final accuracy, but is extremely slow. GraphSAINT-RW only achieves good final accuracy with prohibitively expensive full-batch inference. Node-wise IBMB achieves the best final accuracy with a scalable inference method in 7 out of 10 settings. On ogbn-papers100M, IBMB has a substantially faster time per epoch and lower memory consumption than previous methods demonstrating IBMB's favorable scaling with dataset size. It even performs as well as SIGN-XL ((66.1±0.2) %) [14], while using 30x fewer parameters and no hyperparameter

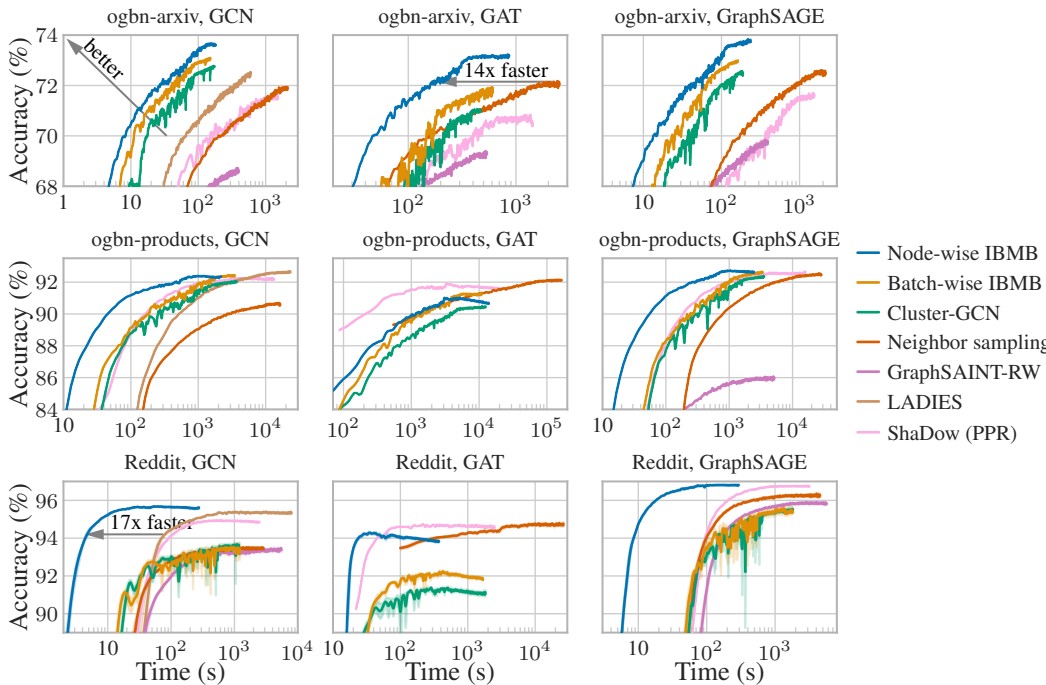

**Figure 3:** Training convergence of validation accuracy in log. time. Average and 95 % confidence interval of 10 runs. GraphSAINT-RW does not reach the shown accuracy range in some settings due to its bad validation performance. IBMB converges the fastest in 9 out of 10 settings.

tuning. Notably, we were unable to evaluate GraphSAINT-RW and Cluster-GCN on this dataset, since they use more than 256 GB of main memory.

**Preprocessing.** IBMB requires more preprocessing than previous methods. However, since IBMB is rather insensitive to hyperparameter choices (see Fig. 5, Table 5), preprocessing rarely needs to be re-run. Instead, its result can be saved to disk and re-used for training different models. Just considering our 10 training seeds, preprocessing of node-wise IBMB only took 1.3 % of the training time for GCN and 0.25 % for GAT on ogbn-arxiv.

**Training set size.** The ogbn-arxiv and ogbn-products datasets both contain a large number of training nodes (91k and 197k, respectively). However, labeling training samples is often an expensive endeavor, and models are commonly trained with only a few hundred or thousand training samples. GraphSAINT-RW and Cluster-GCN are global training methods, i.e. they always use the whole graph for training. They are thus ill-suited for the common setting of a large overall graph containing a small number of training nodes (resulting in a small label rate). In contrast, the training time of IBMB purely scales with the number of training nodes. To demonstrate this, we reduce the label rate by sub-sampling the training nodes of ogbn-products and compare the convergence in Fig. 4. As expected, the gap in convergence speed between IBMB and both Cluster-GCN and GraphSAINT-RW grows even larger for smaller training sets.

**Ablation studies.** We ablate our output node partitioning schemes by instead batching together random sets of nodes. We use fixed batches since we found that resampling incurs significant overhead without benefits – which is consistent with our considerations on gradient samples and contiguous memory accesses. Fig. 6 shows that this method ("Fixed random") converges more slowly and does not reach the same level of accuracy as our partition schemes. Node-wise IBMB converges the fastest, which suggests a trade-off between full gradient samples (random batching) and maximum batch overlap (graph partitioning). Fig. 2 shows that random batching ("IBMB, rand batch.") is also substantially slower and often less accurate for inference. This is due to the synergy effects of output node partitioning: If output nodes have similar auxiliary nodes, they benefit from each other's neighborhood. We ablate auxiliary node selection by comparing IBMB to Cluster-GCN,

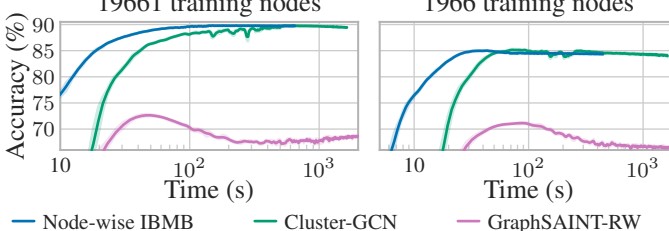
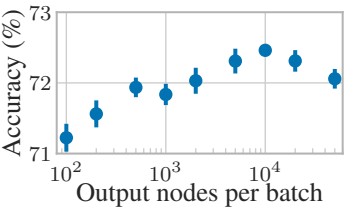

**Figure 4:** Training convergence in log. time for GCN on ogbn-products with smaller training sets. The gap in convergence speed between IBMB and the baselines grows larger for small training sets, since IBMB scales with training set size and not with overall graph size.

**Figure 5:** Trained accuracy for node-wise IBMB, depending on the output nodes per batch (GCN, ogbn-arxiv). The impact of this choice is rather minor.

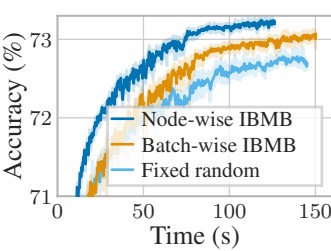
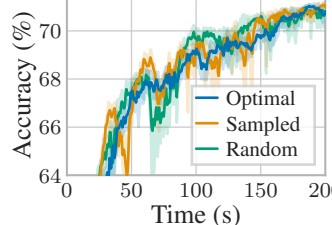
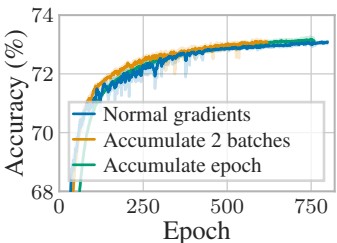

**Figure 6:** Convergence per time for training GCN on ogbn-arxiv. Both batch-wise and node-wise IBMB lead to faster convergence than fixed random batches.

**Figure 7:** Batch scheduling for GAT on ogbn-arxiv. Optimal batch order prevents downward spikes in accuracy and leads to higher final accuracy.

**Figure 8:** Gradient accumulation for batch-wise IBMB on GCN, ogbn-arxiv. The difference is minor, even when accumulating over the full epoch.

since it just uses the graph partition as a batch instead of smartly selecting auxiliary nodes. We use the graph partition size as the number of auxiliary nodes for batch-wise IBMB to allow for a direct comparison. As discussed above, Cluster-GCN consistently performs worse, especially in terms of final accuracy, for inference, and for small label rates. Finally, Fig. 7 compares the proposed batch scheduling methods. Optimal and weighted sampling-based scheduling improve convergence and prevent or reduce downward spikes in accuracy.

**Sensitivity analysis.** Different IBMB is largely insensitive to different local clustering methods and hyperparameters for selecting auxiliary nodes (see Table 5). Even increasing the number of output nodes per batch with a fixed number of auxiliary nodes per output node only has a minor impact on accuracy, especially above 1000 output nodes per batch, as shown by Fig. 5. IBMB performs well even in extremely constrained settings with small batches of 100 output nodes per batch. In practice, IBMB only has one free hyperparameter: The number of auxiliary nodes per output node, which allows optimizing for accuracy or speed. The number of output nodes per batch is then given by the available GPU memory, while the local clustering method and other hyperparameters are not important.

**Gradient accumulation.** Accumulating gradients across multiple batches is a method for smoothing batches if gradient noise is too high. We might expect this to happen in IBMB due to the sparse gradients caused by primary node partitioning. However, Fig. 8 shows that gradient accumulation in fact only has an insignificant effect on IBMB, demonstrating its stability during training.

# 6   Conclusion

We propose influence-based mini-batching (IBMB), a method for extracting batches for GNNs. IBMB formalizes creating batches for inference by maximizing the influence score on the output nodes. Remarkably, with an adaptive optimizer and batch scheduling IBMB even performs well

during training. It improves training convergence by up to 17x and inference time by up to 130x compared to previous methods that reach similar accuracy. IBMB performs especially well for sparse labels, large datasets, and when the pipeline is constrained by data access speed.

## Acknowledgments

This work was supported by the Deutsche Forschungsgemeinschaft (DFG) through TUM International Graduate School of Science and Engineering (IGSSE) (GSC 81).

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

# A   Proof of Theorem 1

**Path-based view of neural networks.** We can view a neural network with ReLUs as a directed acyclic computational graph and express the $i$'th output logit via paths through this graph as

$$\boldsymbol{h}_i^{(L)} = \frac{1}{\lambda^{(H-1)/2}} \sum_{q=1}^{\phi} Z_{i,q} X_{i,q} \prod_{l=1}^{L} w_{i,q}^{(l)}, \tag{8}$$

where $\lambda$ is a constant related to the size of the network [8] and $\phi$ is the total number of paths. Furthermore, $Z_{(i,q)} \in \{0,1\}$ denotes whether the path $q$ is active or inactive when any ReLU is deactivated. $X_{(i,q)}$ represents the input feature used in the $q$-th path of logit $i$, and $w_{(i,q)}^{(l)}$ the used entry of the weight matrix $W_l$ in layer $l$.

**Path-based view of GNNs.** We can extend this framework to graph neural networks by additionally introducing paths $p$ through the (data-based) graph, starting from the auxiliary node $v$ and ending at the output node $u$, as

$$\boldsymbol{h}_{u,i}^{(L)} = \frac{1}{\lambda^{(H-1)/2}} \sum_{v \in \mathcal{V}} \sum_{p=1}^{\psi} \sum_{q=1}^{\phi} Z_{v,p,i,q} X_{v,p,i,q} \prod_{l=1}^{L} a_{v,p}^{(l)} w_{i,q}^{(l)}, \tag{9}$$

where $\psi$ is the total number of graph-based paths and $a_{v,p}^{(l)}$ denotes the graph-dependent but feature-independent aggregation weights. Note that $a_{v,p}^{(l)}$ depends on the whole path $(v,p)$ and can thus be a function of any node and edge on this path, including the current and next layer's nodes.

**Expected influence score.** To obtain the influence score, we calculate the derivative

$$\frac{\partial \boldsymbol{h}_{u,i}^{(L)}}{\partial X_{v,j}} = \frac{1}{\lambda^{(H-1)/2}} \sum_{p=1}^{\psi} \sum_{q=1}^{\phi'} Z_{v,p,i,q} \prod_{l=1}^{L} a_{v,p}^{(l)} w_{i,q}^{(l)}, \tag{10}$$

with $X_{v,j}$ denoting input feature $j$ at node $v$ and $\phi'$ denoting the number of computational paths with input feature $j$. To simplify this expression, we use the assumption that all paths $(v,p,i,q)$ are activated with the same probability $\rho$, i.e. $\mathbb{E}[Z_{v,p,i,q}] = \rho$, and compute the expectation:

$$
\begin{aligned}
\mathbb{E}\left[\frac{\partial \boldsymbol{h}_{u,i}^{(L)}}{\partial X_{v,j}}\right] &= \frac{1}{\lambda^{(H-1)/2}} \sum_{p=1}^{\psi} \sum_{q=1}^{\phi'} \rho \prod_{l=1}^{L} a_{v,p}^{(l)} w_{i,q}^{(l)} \\
&= \frac{\rho}{\lambda^{(H-1)/2}} \left( \sum_{p=1}^{\psi} \prod_{l=1}^{L} a_{v,p}^{(l)} \right) \left( \sum_{q=1}^{\phi'} \prod_{l=1}^{L} w_{i,q}^{(l)} \right).
\end{aligned}
\tag{11}
$$

The only node-dependent term in the expected influence score $\mathbb{E}[I(v,u)] = \sum_i \sum_j \mathbb{E}\left[\left|\frac{\partial \boldsymbol{h}_{u,i}^{(L)}}{\partial \boldsymbol{X}_{v,j}}\right|\right]$ is thus $\left|\sum_{p=1}^{\psi} \prod_{l=1}^{L} a_{v,p}^{(l)}\right|$.

**Expected output.** We similarly obtain the expected output by additionally using the assumption that features have a node-independent expected value $\mathbb{E}[X_{v,p,i,q}] = \chi_{i,q}$, yielding

$$
\begin{aligned}
\mathbb{E}[\boldsymbol{h}_{u,i}^{(L)}] &= \frac{1}{\lambda^{(H-1)/2}} \sum_{v \in \mathcal{V}} \sum_{p=1}^{\psi} \sum_{q=1}^{\phi} \rho \chi_{i,q} \prod_{l=1}^{L} a_{v,p}^{(l)} w_{i,q}^{(l)} \\
&= \frac{\rho}{\lambda^{(H-1)/2}} \sum_{v \in \mathcal{V}} \left( \sum_{p=1}^{\psi} \prod_{l=1}^{L} a_{v,p}^{(l)} \right) \left( \sum_{q=1}^{\phi} \chi_{i,q} \prod_{l=1}^{L} w_{i,q}^{(l)} \right).
\end{aligned}
\tag{12}
$$

Again, the only node-dependent term in the expected output is $\sum_{p=1}^{\psi} \prod_{l=1}^{L} a_{v,p}^{(l)}$. Adding any input node thus changes node $u$'s output in absolute terms by

$$\left| \sum_{p=1}^{\psi} \prod_{l=1}^{L} a_{v,p}^{(l)} \right| C = \mathbb{E}[I(v,u)] C', \tag{13}$$

with $C$ and $C'$ denoting all node-independent terms. Selecting the input nodes with maximum influence score $I(v,u)$ thus minimizes the $L_1$ norm of the approximation error. Note that this choice only considers the effect of selecting input nodes. It does not model the effect of changing the graph.

# B  Model and training details

**Hardware.** All experiments are run on an NVIDIA GeForce GTX 1080Ti. The experiments on ogbn-arxiv and ogbn-products use up to 64 GB of main memory. The experiments on ogbn-papers100M use up to 256 GB.

**Packages.** Our experiments are based on the following packages and versions:

- torch-geometric 1.7.0
    - torch-cluster 1.5.9
    - torch-scatter 2.0.6
    - torch-sparse 0.6.9
- python 3.7.10
- ogb 1.3.1
- torch 1.8.1
- cudatoolkit 10.2.89
- numba 0.53.1
- python-tsp 0.2.0

**Preprocessing.** Before training, we first make the graph undirected, and add self-loops. The adjacency matrix is symmetrically normalized. We cache the symmetric adjacency matrix for graph partitioning and mini-batching. Instead of re-calculating the adjacency matrix normalization factors for GCN for each mini-batch, we re-use the global normalization factors. We found this to achieve similar accuracy at lower computational cost.

**Models.** We use three models for all the experiments: GCN (3 layers, hidden size 256 for the ogbn datasets and 2 layers, hidden size 512 for Reddit), GAT (3 layers, hidden size 128, 4 heads for the ogbn datasets and 2 layers, hidden size 64, 4 heads for Reddit), and GraphSAGE (3 layers, hidden size 256). All models use layer normalization, ReLU activation functions, and dropout. We performed a grid search on ogbn-arxiv, ogbn-products, and Reddit to obtain the optimal model hyperparameters based on final validation accuracy. For ogbn-papers100M we use the same hyperparameters as for GCN on ogbn-arxiv, but with 32 auxiliary nodes per output node.

**Training.** We use the Adam optimizer for all experiments, with a starting learning rate of $10^{-3}$. We use an $L_2$ regularization of $10^{-4}$ for GCN on ogbn-arxiv and ogbn-products, and no $L_2$ regularization in all other settings. We use a ReduceLROnPlateau scheduler for the optimizer, with the decay factor 0.33, patience 30, minimum learning rate $10^{-4}$, and cooldown of 10, based on validation loss. We train for 300 to 800 epochs and stop early with a patience of 100 epochs, based on validation loss. We determine the optimal batch order for IBMB via simulated annealing [11].

**Training for inference.** For comparing inference performance in Fig. 2 we trained 10 separate models per setting with node-wise IBMB. We then used the same 10 models to evaluate every method. Fig. 9 shows that the choice of training method does not impact our findings. The reasons for this are that the training method (1) does not influence inference time and (2) is not able to bridge the large accuracy disadvantages that some methods have.

**Batch-wise IBMB.** We tune the number of batches and thus the size of batches using a grid search (see Table 1). Generally, final accuracy increases with larger batch sizes, but this can lead to excessive memory usage and slower convergence speed. The resulting partitions then define the output nodes in each batch. We use as many auxiliary nodes as the size of each partition. However, the auxiliary nodes will be different than the partition since they are selected based on the output nodes via batch-wise clustering. Note that the inference batch size is double the sizes of training batches since in this case we do not need to store any gradients.

**Node-wise IBMB.** For node-wise batching we first calculate the PPR scores for each output node, and then pick the top-k nodes as its auxiliary nodes. Generally we use the same batch size, i.e. number of nodes in a batch, as in batch-wise IBMB, to keep the GPU memory usage similar. However, if the graph is too dense, we might have to increase the batch size of node-wise IBMB, because it tends to create sparser batches. We tune the number of auxiliary nodes per output node using a logarithmic grid search using factors of 2. Based on this we use 16 neighbors for ogbn-arxiv, 64 for ogbn-products, 8 for Reddit, and 96 for ogbn-papers100M. Note that the number of auxiliary nodes is the main degree of freedom in IBMB. It influences preprocessing time, runtime, memory usage, and accuracy. The number of output nodes per batch is then determined by the available GPU memory.

**Table 1:** Number of batches for batch-wise IBMB.

| Model | Dataset | Number of batches | | |
|---|---|---|---|---|
| | | Train | Validation | Test |
| GCN | ogbn-arxiv | 4 | 2 | 2 |
| GCN | ogbn-products | 16 | 8 | 8 |
| GCN | Reddit | 8 | 4 | 4 |
| GAT | ogbn-arxiv | 8 | 4 | 4 |
| GAT | ogbn-products | 1024 | 512 | 512 |
| GAT | Reddit | 400 | 200 | 200 |
| GCN | ogbn-papers100M | 256 | 32 | 48 |

**Table 2:** Hyperparameters for LADIES

| Model | Dataset | Nodes per layer | |
|---|---|---|---|
| | | Train | Validation |
| GCN | ogbn-arxiv | 42 336 | 84 672 |
| GCN | ogbn-products | 204 085 | 306 128 |
| GCN | Reddit | 90 000 | 150 000 |

**Approximate PPR.** Calculating the full personalized PageRank (PPR) matrix is prohibitively expensive for large graphs. To enable fast preprocessing times, we approximate node-wise PPR using a push-flow algorithm [2] with a fixed number of iterations and approximate batch-wise PPR using power iterations. Both variants are based on parallel sparse matrix operations on GPU. We choose their hyperparameters so they do not impede accuracy while still having a reasonable preprocessing time. We use 50 power iterations for batch-wise PPR. For node-wise PPR we use three iterations, $\epsilon = 0.0002$ for ogbn-arxiv, $\epsilon = 0.0005$ for ogbn-products, and $\epsilon = 0.000\,02$ for Reddit and ogbn-papers100M. For node-wise PPR we additionally downsample the unusually dense Reddit adjacency matrix to an average of 8 neighbors per node.

**Random batching.** Random batching is similar to node-wise IBMB except that the auxiliary nodes are batched randomly. We first calculate the PPR scores and pick the top-k neighbors as the auxiliary nodes for a output node. We choose the same number of neighbors as with node-wise IBMB. We investigate 2 variants of random batching: Resampling the batches in every epoch, and sampling them once during preprocessing and then fixing the batches. We only show the results for the second method, since we found it to be significantly faster, albeit requiring significantly more main memory.

**Hyperparameter tuning.** The priorities for tuning the hyperparameters are as follows: 1. To keep methods comparable in a realistic setup, we keep the GPU memory usage constant between methods. 2. When there are semantic hyperparameters that do not influence performance (such as the number of steps per epoch in GraphSAINT-RW, which only changes how an epoch is defined), we choose them to be comparable to other methods. 3. We choose all relevant hyperparameters based on validation accuracy. If a hyperparameter is not critical to memory usage we tune it per dataset and not per model. We use this process for both IBMB and the baselines.

**Baseline hyperparameters.** For Cluster-GCN the number of batches are the same as for batch-wise IBMB. Table 2 shows the hyperparameters for LADIES, Table 3 for neighbor sampling, and Table 4

**Table 3:** Hyperparameters for neighbor sampling

| Model | Dataset | Number of batches | | | Number of nodes |
|---|---|---|---|---|---|
| | | Train | Validation | Test | |
| GCN | ogbn-arxiv | 12 | 8 | 8 | 6, 5, 5 |
| GCN | ogbn-products | 20 | 4 | 200 | 5, 5, 5 |
| GCN | Reddit | 8 | 4 | 4 | 12, 12 |
| GAT | ogbn-arxiv | 8 | 4 | 4 | 8, 7, 5 |
| GAT | ogbn-products | 1000 | 150 | 8000 | 15, 10, 10 |
| GAT | Reddit | 400 | 400 | 400 | 20, 20 |

**Table 4:** Hyperparameters for GraphSAINT-RW

| | | | | | Batch size | |
|---|---|---|---|---|---|---|
| Model | Dataset | Walk length | Sample coverage | Number of steps | Train | Val/Test |
| GCN | ogbn-arxiv | 2 | 100 | 4 | 25 000 | 10 000 |
| GCN | ogbn-products | 2 | 100 | 16 | 80 000 | 5000 |
| GCN | Reddit | 2 | 100 | 8 | 23 000 | 6000 |
| GAT | ogbn-arxiv | 2 | 100 | 8 | 17 500 | 10 000 |
| GAT | ogbn-products | 2 | 100 | 1024 | 14 000 | 100 |
| GAT | Reddit | 2 | 100 | 400 | 1600 | 60 |

**Table 5:** Methods and hyperparameters for selecting auxiliary nodes for GCN on ogbn-products with batch-wise IBMB. IBMB is very robust to this choice. We did observe a slightly lower validation accuracy for low alpha (0.05). We always use 0.25.

| | | Time (s) | Test accuracy (%) | |
|---|---|---|---|---|
| Method | $\alpha, t$ | per epoch | IBMB inference | Full-batch |
| PPR | 0.05 | 3.5 | 76.8±0.3 | 77.1±0.3 |
| PPR | 0.15 | 3.6 | 76.6±0.4 | 76.9±0.4 |
| PPR | 0.25 | 3.5 | 76.8±0.2 | 77.2±0.3 |
| PPR | 0.35 | 3.5 | 76.9±0.5 | 77.2±0.5 |
| Heat kernel | 0.1 | 3.5 | 76.5±0.4 | 76.8±0.3 |
| Heat kernel | 1 | 3.5 | 76.6±0.5 | 76.9±0.5 |
| Heat kernel | 3 | 3.5 | 76.8±0.2 | 77.1±0.2 |
| Heat kernel | 5 | 3.5 | 76.7±0.5 | 77.0±0.5 |
| Heat kernel | 7 | 3.5 | 76.6±0.4 | 76.8±0.4 |

for GraphSAINT-RW. To ensure that every node is visited exactly once during GraphSAINT-RW inference we use the validation/test nodes only as root nodes of the random walks.

**Full-batch inference.** We chunk the adjacency matrix and feature matrix for full-batch inference to allow using the GPU even for larger datasets. The only hyperparameter is the number of chunks. We limit the chunk size to ensure that full-batch inference does not exceed the amount of GPU memory used during training.

**Experimental limitations.** We only tested our method on homophilic node classification datasets. While proximity is a central inductive bias in all GNNs, we did not explicitly test this on a more general variety of graphs. However, note that IBMB does *not* require homophily. The underlying assumption is merely that nearby nodes are the most important, not that they are similar. Finally, we expect our method to perform even better in the context of billion-node graphs, but our benchmark datasets still fit into main memory.

## C   Ethical considerations

Scalable graph-based methods can enable the fast analysis of huge datasets with billions of nodes. While this has many positive use cases, it also has obvious negative repercussions. It can enable mass surveillance and the real-time analysis of whole populations and their social networks. This can potentially be used to detect emerging resistance networks in totalitarian regimes, thus suppressing chances for positive change. Voting behavior is another typical application of network analysis: Voters of the same party are likely to be connected to one another. Scalable GNNs can thus influence voting outcomes if they are leveraged for targeted advertising.

The ability of analyzing whole populations can also have negative personal effects in fully democratic countries. If companies determine credit ratings or college admission based on connected personal data, a person will be even more determined by their environment than they already are. Companies might even leverage the obscurity of complex GNNs to escape accountability: It might be easy to reveal the societal effects of your housing district, but unraveling the combined effects of your social networks and digitally connected behavior seems almost impossible. Scalable GNNs might thus make it even more difficult for individuals to escape the attractive forces of the status quo.

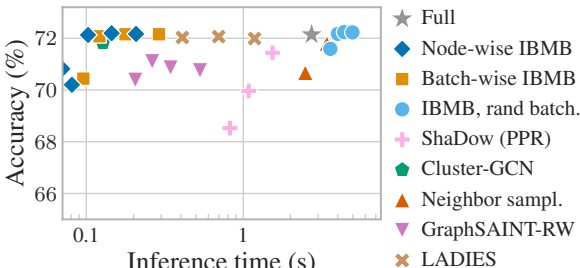

**Figure 9:** Test accuracy and log. inference time on ogbn-arxiv for 10 GCNs trained with GraphSAINT-RW. The pretraining method does have an impact on the accuracy of some methods, but not enough to change any experimental findings.

**Table 6:** Main memory usage (GiB). In some settings, IBMB uses more main memory than previous methods due to overlapping batches (e.g. on ogbn-products). However, it can also reduce memory requirements because it ignores irrelevant parts of the graph (e.g. on Reddit). Note that our hyperparameters keep GPU memory usage consistent between methods, as opposed to main memory usage.

| | ogbn-arxiv | | | ogbn-products | | | Reddit | | |
|---|---|---|---|---|---|---|---|---|---|
| | GCN | GAT | GraphSAGE | GCN | GAT | GraphSAGE | GCN | GAT | GraphSAGE |
| Neighbor sampling | 3.0 | 3.6 | 3.1 | 8.7 | 7.9 | 8.5 | 7.4 | 7.5 | 7.1 |
| LADIES | 3.0 | - | - | 6.0 | - | - | 4.8 | - | - |
| GraphSAINT-RW | 3.5 | 3.6 | 3.5 | 9.6 | 9.6 | 9.6 | 8.4 | 8.5 | 8.4 |
| Cluster-GCN | 3.5 | 3.4 | 3.5 | 7.8 | 6.0 | 7.3 | 6.1 | 4.2 | 6.5 |
| Batch-wise IBMB | 3.5 | 3.6 | 3.5 | 7.9 | 7.0 | 7.8 | 6.3 | 4.9 | 6.3 |
| Node-wise IBMB | 3.8 | 3.8 | 4.2 | 13.0 | 12.3 | 13.2 | 4.5 | 5.3 | 5.1 |

## D   Additional results

**Main memory usage.** IBMB's main memory usage depends on three aspects: 1. How large is the training/validation set compared to the full graph? 2. How many auxiliary nodes per output node are we using? 3. How well are the auxiliary nodes overlapping per batch? As shown in Table 6, IBMB increases main memory usage in some settings, which is due to the overlap between batches. However, in other settings it reduces memory requirements because it ignores irrelevant parts of the graph and removes the dataset from memory after preprocessing. Note that our hyperparameters keep *GPU* memory usage consistent between methods.

**Table 7:** Final accuracy and runtime averaged over 10 runs, with standard deviation. "Same method" refers to using the training method for inference, while "full-batch" uses the whole graph for inference. IBMB achieves similar accuracy as previous methods when used for training, while using significantly less time per epoch and without requiring full-batch inference. IBMB is up to 900x faster (ogbn-papers100M) than using full-batch inference, at comparable accuracy. Other inference methods are substantially slower or less accurate. Note that LADIES is incompatible with the self loops in GAT and GraphSAGE.

| Setting | Training method | Time (s) | | | Test accuracy (%) | |
|---|---|---|---|---|---|---|
| | | Preprocess | Per epoch | Inference | Same method | Full-batch |
| ogbn-arxiv, GCN | Full-batch | - | - | 2.8 | - | - |
| | Neighbor sampling | **0.3** | 4.7 | 2.5 | 70.7±0.2 | 71.3±0.4 |
| | LADIES | **0.3** | 0.62 | 0.69 | 71.7±0.2 | 71.4±0.3 |
| | GraphSAINT-RW | 0.4 | 0.42 | 0.34 | 68.1±0.2 | *72.3±0.2* |
| | ShaDow (PPR) | 8.3 | 2.69 | 1.40 | 70.9±0.2 | 72.0±0.1 |
| | Cluster-GCN | 8.7 | **0.14** | *0.14* | 72.0±0.1 | *72.2±0.1* |
| | **Batch-wise IBMB** | 14.1 | **0.14** | **0.13** | *72.2±0.2* | *72.2±0.2* |
| | **Node-wise IBMB** | 17.5 | 0.27 | 0.16 | **72.6±0.1** | **72.6±0.1** |
| ogbn-arxiv, GAT | Full-batch | - | - | 9.4 | - | - |
| | Neighbor sampling | **0.3** | 4.1 | 1.97 | *70.9±0.1* | *72.1±0.1* |
| | GraphSAINT-RW | *0.4* | 1.2 | 0.38 | 68.7±0.2 | **72.6±0.1** |
| | ShaDow (PPR) | 8.4 | 2.98 | 1.30 | 70.3±0.2 | 71.8±0.1 |
| | Cluster-GCN | 7.6 | *0.69* | **0.28** | 69.7±0.3 | 71.6±0.2 |
| | **Batch-wise IBMB** | 7.7 | **0.68** | *0.31* | *71.0±0.3* | 71.8±0.3 |
| | **Node-wise IBMB** | 17.6 | 1.52 | 0.93 | **72.0±0.2** | *72.2±0.2* |
| ogbn-arxiv, GraphSAGE | Full-batch | - | - | 2.37 | - | - |
| | Neighbor sampling | **0.3** | 3.44 | 1.67 | 71.1±0.1 | 72.0±0.1 |
| | GraphSAINT-RW | **0.3** | 0.41 | 0.35 | 69.0±0.1 | *72.2±0.1* |
| | ShaDow (PPR) | 7.7 | 2.68 | 1.38 | 70.9±0.2 | 71.9±0.2 |
| | Cluster-GCN | 8.8 | **0.15** | *0.14* | 71.7±0.1 | 72.1±0.1 |
| | **Batch-wise IBMB** | 7.2 | **0.15** | **0.13** | *72.0±0.2* | 72.1±0.1 |
| | **Node-wise IBMB** | 17.5 | 0.31 | 0.16 | **72.4±0.2** | **72.4±0.1** |
| ogbn-products, GCN | Full-batch | - | - | 130 | - | - |
| | Neighbor sampling | **32** | 42 | 433 | **78.2±0.2** | 78.0±0.2 |
| | LADIES | *33* | 25 | 22.5 | 75.9±0.3 | *79.0±0.4* |
| | GraphSAINT-RW | 35 | 11 | 20.4 | 53.6±0.6 | **79.9±0.2** |
| | ShaDow (PPR) | 299 | 28.5 | 1242 | *77.5±0.3* | 77.4±0.4 |
| | Cluster-GCN | 302 | *3.7* | *3.4* | 76.2±0.3 | 76.5±0.2 |
| | **Batch-wise IBMB** | 306 | **3.5** | **3.1** | 76.8±0.2 | 77.2±0.3 |
| | **Node-wise IBMB** | 382 | 5.4 | 13.8 | 77.3±0.3 | 77.3±0.3 |
| ogbn-products, GAT | Full-batch | - | - | 1700 | - | - |
| | Neighbor sampling | **33** | 450 | 3450 | **79.1±0.3** | 77.2±0.5 |
| | GraphSAINT-RW | *35* | 140 | 102 | 69.5±0.1 | **80.8±0.2** |
| | ShaDow (PPR) | 298 | 66 | 1461 | 77.3±0.3 | *79.2±0.4* |
| | Cluster-GCN | 626 | **24** | *10.6* | 76.6±0.4 | 78.1±0.5 |
| | **Batch-wise IBMB** | 767 | *25* | **10.0** | 77.0±0.4 | *78.9±0.6* |
| | **Node-wise IBMB** | 378 | 42 | 97 | **78.9±0.3** | 79.0±0.3 |
| ogbn-products, GraphSAGE | Full-batch | - | - | 88.0 | - | - |
| | Neighbor sampling | **31.4** | 52.0 | 530 | **81.0±0.2** | **81.4±0.2** |
| | GraphSAINT-RW | *35.8* | 10.6 | 20.0 | 69.4±0.2 | **81.3±0.2** |
| | ShaDow (PPR) | 298 | 28.3 | 1169 | 80.0±0.3 | 80.8±0.3 |
| | Cluster-GCN | 313 | *3.1* | *3.4* | 79.5±0.4 | 79.7±0.4 |
| | **Batch-wise IBMB** | 319 | **2.9** | **3.1** | 79.2±0.3 | 79.5±0.3 |
| | **Node-wise IBMB** | 374 | 5.1 | 13.3 | *80.6±0.3* | 80.8±0.3 |

*Continued on the next page.*

Final accuracy and runtime averaged over 10 runs, continued.

| Setting | Training method | Time (s) | | | Test accuracy (%) | |
|---|---|---|---|---|---|---|
| | | Preprocess | Per epoch | Inference | Same method | Full-batch |
| Reddit, GCN | Full-batch | - | - | 14.8 | - | - |
| | Neighbor sampling | **14.4** | 7.3 | 3.3 | 93.5±0.1 | 94.8±0.1 |
| | LADIES | *15.4* | 11.4 | 11.4 | *95.5±0.0* | *95.3±0.0* |
| | GraphSAINT-RW | 17.1 | 14.6 | 2.9 | 93.2±0.1 | **95.6±0.0** |
| | ShaDow (PPR) | 54.0 | 7.4 | 2.2 | 95.2±0.1 | 95.0±0.0 |
| | Cluster-GCN | 175 | 1.8 | 1.6 | 93.7±0.2 | 94.8±0.1 |
| | **Batch-wise IBMB** | 175 | *1.6* | *1.4* | 93.5±0.4 | 94.7±0.1 |
| | **Node-wise IBMB** | 64.8 | **0.74** | **0.59** | **95.7±0.1** | *95.2±0.1* |
| Reddit, GAT | Full-batch | - | - | 76.9 | - | - |
| | Neighbor sampling | **14.8** | 70 | 32.5 | *94.3±0.1* | *95.1±0.1* |
| | GraphSAINT-RW | *17.9* | 21 | 3.2 | 79.4±0.2 | **95.4±0.1** |
| | ShaDow (PPR) | 56.5 | 7.0 | 1.7 | **94.6±0.1** | 94.1±0.2 |
| | Cluster-GCN | 366 | 4.7 | 1.4 | 91.4±0.1 | 93.5±0.7 |
| | **Batch-wise IBMB** | 396 | *4.3* | *1.2* | 91.6±0.1 | 92.8±1.1 |
| | **Node-wise IBMB** | 65.3 | **1.1** | **0.25** | *94.2±0.1* | 94.1±0.3 |
| Reddit, GraphSAGE | Full-batch | - | - | 17.3 | - | - |
| | Neighbor sampling | **16.1** | 7.5 | 3.5 | 96.2±0.0 | **96.8±0.0** |
| | GraphSAINT-RW | *18.2* | 14.6 | 3.6 | 95.9±0.0 | **96.8±0.0** |
| | ShaDow (PPR) | 56.5 | 7.3 | 2.4 | **96.8±0.0** | 96.4±0.0 |
| | Cluster-GCN | 173 | 1.7 | 1.8 | 95.5±0.2 | 96.0±0.1 |
| | **Batch-wise IBMB** | 175 | *1.6* | *1.7* | 95.6±0.2 | 96.1±0.1 |
| | **Node-wise IBMB** | 66.0 | **0.78** | **0.65** | **96.8±0.0** | 96.5±0.0 |
| papers 100M, GCN | Full-batch | - | - | 5700 | - | - |
| | Neighbor sampling | 739 | *900* | *159* | 64.3±0.2 | 61.8±0.2 |
| | LADIES | 735 | 2830 | 672 | *65.4±0.2* | *62.4±0.4* |
| | **Node-wise IBMB** | 2290 | **51** | **6.2** | **66.1±0.1** | **66.0±0.1** |

