# OpenReview forum: "Influence-Based Mini-Batching for Graph Neural Networks"
_logconference.io/LOG/2022/Conference — LoG 2022 Oral_

### Official Review · Reviewer_Wndf · 2022-10-14

**Overall Score:** 6
**Confidence:** 3

**Review:**

This paper proposes a mini-batching approach for training GNNs with a focus on model behavior during inference. The proposed mini-batching is based on the influence score of nodes on the outputs. The influence scores between nodes are approximated with Personalized PageRank.

The experimental results show the proposed mini-batching accelerates inference by up to 130X compared to baselines. Even though the approach is based on model behavior during inference, it achieves 18X faster training time. The proposed method also reaches similar accuracy in a faster time.

I recommend accepting this paper as the proposed method is backed up by theoretical understanding and is also approximated to the well-known Personalized PageRank. The results are also encouraging for training with the proposed mini-batching.

---

### Official Review · Reviewer_6vgx · 2022-10-19

**Overall Score:** 10
**Confidence:** 4

**Review:**

Summary Of the Paper:

The paper proposed a method of influence-based mini-batching (IBMB) to mitigate the neighboring sampling or graph clustering overhead due to the expensive random data accesses. This method accelerates inference by up to 130x compared to previous methods that reach similar accuracy, and the experimental results are up to 18x faster training per epoch and up to 17x faster convergence per runtime compared to previous methods.

Overall, I think the strengths of this work are significant in the practical scenarios of using GNNs. However, there are some weaknesses and questions that need to be solved and answered. Hence, I give this work “Weak Accept”. If authors reasonably solve these weaknesses and answer such questions, I will definitely raise the overall score. Here are the strengths, weaknesses, and questions of this work:

Strengths of the paper:

(1). This work proposed a novel, non-random, and locality-based sampling to challenge the representation learning on large graphs, and IBMB is significant and promising thereon.

(2). The experiments show that the IBMB method substantially accelerates inference and training without sacrificing accuracy.

(3). The sentences of this paper are well-written.

Weaknesses of the paper:

(1). Theorem 1 is confusing for me. I don't think Theorem 1 is necessary, which makes this work too mathematical (Over-theorization) and complex to understand, hindering the spread of the work.

(2). The description of how to use Personalized PageRank (PPR) to compute the influence scores is unclear. The authors should provide more details to describe this part, especially from the view of motivation and insight.

(3). Minor grammar or expression errors:

Line 42: On the practical side, we propose……

Line 64: $H$ and $H_e$ are not defined.

Line 65: We denote the embedding in layer……

And some other errors……

Questions of the paper:

(1). I am confused about how PPR defines the influence scores.

(2). What is the theoretical relationship between PPR and Theorem 1?


--------------------------------- new comments -------------------------------------

The authors answered my questions sufficiently, so I decided to raise the overall score of this work. In summary, it is my opinion that this work is significant to the whole GNN community, especially in practical GNN training and inference. Hence, I appreciate this work.

---

### Official Review · Reviewer_ob1G · 2022-10-22

**Overall Score:** 8
**Confidence:** 4

**Review:**

This paper studies the problem of the efficient inference and training for GNN models on large-scale graphs. They propose an efficient mini-batch construction mechanism with graph partitioning and auxiliary nodes sampling based on influence scores, which are defined as the local sensitivity of the output of a node with respect to other input nodes. They theoretically show that the influenced score-based mini-batching construction achieves minimal approximation error for the GNN embeddings under some simplified conditions. With the implementation of personalized PageRank (PPR) for influence score, they conducted extensive experiments on three large-scale datasets over three GNN models to demonstrate the effectiveness of their solution.

## Rating

Overall, this work is nicely done, and the writing is excellent. My major concern is the novelty and performance gap compared with some critical missing references [1]. Therefore I gave this work a boarderline score. If this work comes out earlier, it will be a clear acceptance to me.

## Pros

* The proposed problem of efficient inference and training for GNN on large-scale graphs is critical to apply GNN models in the industry. Their proposed solution seems to be sound and practical.
* Although they introduce some simplification conditions, the theoretical analysis of the influence score is very interesting and inspiring.
* The experiment results are very strong and solid. They did extensive hyperparameter tuning for the other baselines. Besides, they provided open-sourced code.
* The paper is well-written and easy to follow.

## Concerns

* Missing comparison against some important references, e.g., [1] [2]. One of the major advantages of their algorithm comes from the decoupling of label nodes sampling and the auxiliary nodes sampling. [1] introduces a framework covering broader aspects, while this work introduced an effective implementation using PPR to construct the mini-batching.
* The primary claim is efficient inference, and efficient training sounds like a side product. However, most of the parts talk about the training aspect. I did not see why the method is specifically advantageous on the inference side against the training. It looks more natural if the paper focuses on the training side.
* The results in Figure 2 are mainly from the efficient training work. There are no other baselines specifically tailored to efficient inference, e.g., [3].

## Minor issues

* Fig. 2 could be more readable with different marker shapes.
* Fig. 5: does this fluctuation of 2% really means “in-sensitive”?

## Questions
* For the results in Figure. 2, does it include the prepossessing time?
* Line 268: all results are based on the same pre-trained model. What method was applied to get the pre-trained model? Does this have an impact on the result?
* Line 285: why graphSAINT needs a full batch inference? The trivial solution is to use the same sampling method from the training to make the inference.
* Fig. 3, why the node-wise IBMB runs much faster than clusterGCN for each mini-batch? I understand that the convergence of node-wise IBMB could be better than cluster-GCN since it takes better auxiliary nodes. However, in terms of running complexity for each minibatch, I did not see why node-wise IBMB is much faster than cluserGCN.

## References

1. Zeng, Hanqing, et al. "Decoupling the Depth and Scope of Graph Neural Networks." Advances in Neural Information Processing Systems. 2021.
1.  https://docs.dgl.ai/generated/dgl.dataloading.ShaDowKHopSampler.html#dgl.dataloading.ShaDowKHopSampler

1. Zhang, Shichang, et al. "Graph-less neural networks: Teaching old mlps new tricks via distillation." arXiv preprint arXiv:2110.08727 (2021).

---

### Official Review · Reviewer_rqmi · 2022-10-25

**Overall Score:** 8
**Confidence:** 4

**Review:**

Summary of the paper:

The paper proposes a theoretical framework called Influence-based mini-batching (IBMB) for selecting mini-batches in GNN inferencing phase. The framework can be adopted with various state of the art GNN architecture such as GCN, GraphSage etc. Through extensive experiments, the paper shows that their framework achieves orders of magnitude speed up without sacrificing accuracy.

Strengths:

1. The proposed optimization problem (eq 5 and 6) for the IBMB problem is interesting, novel and backed by reasonable theoretical arguments.

2. The framework part of the paper is generally well presented, easy to follow, and detailed.

3. The experiments are thorough and supports the efficiency of the proposed framework.


Weaknesses:

1. Some aspects of the framework is not very clear from the reading. Please see my question below.

2. The experiments section can be a bit more organized.


Recommendation:

Overall, I like the paper. I believe the contribution of the work is novel and significant. I recommend weak accept for this paper.

Questions and Suggestions:

Try to incorporate the following in the presentation of your work.

1. In the definition of influence scores (equation 3), X is not defined.

2. On line 99-101, it says the score is useful when the model and the node features are not available. However, this contradicts the definition of the score since it depends on the embedding and the input feature.

3. Why is PPR a reasonable approximation for the influence score function (eq 3). A discussion on the same is required.

4. On line 185, its discussed that METIS can be used to partition the output nodes. Does all of the edges of the graphs are used during this partitioning? Additionally, METIS roughly partitions the graph, minimizing cut edges. This might lead to imbalanced output nodes in different partitions. Does this not pose an issue?

5. In the computational advantages section at line 199, the authors discuss caching of the mini-batches. While its true that selecting auxiliary nodes at random in each epoch is more expensive in terms of computations, but they do seem to generalize well. Is there a concern regarding this with fixed mini-batches?

6. In Fig 2, IBMB,randbatch is plotted, but it was not introduced in the experiments section.

7. A subsection in the Experiments section should be added highlighting the major findings of the experiments sections.

=================================================================================================

Based on the reviewer response, I decided to increase my score from weak accept to clear accept.

---

### Meta-Review · Area_Chair_USjS · 2022-11-09

**Confidence:** 3
**Recommendation:** Accept

**Meta Review:**

All reviewers agreed this paper is worth publishing.

---

### Decision · Program_Chairs · 2022-11-23

Accept (Oral)